# Ferulic Acid Induces Keratin 6α via Inhibition of Nuclear β-Catenin Accumulation and Activation of Nrf2 in Wound-Induced Inflammation

**DOI:** 10.3390/biomedicines9050459

**Published:** 2021-04-22

**Authors:** Kang-Hoon Kim, Ji Hoon Jung, Won-Seok Chung, Chang-Hun Lee, Hyeung-Jin Jang

**Affiliations:** 1Department of Science in Korean Medicine, Graduate School, Kyung Hee University, 26 Kyungheedae-ro, Dongdaemun-gu, Seoul 02447, Korea; poklmoo@naver.com (K.-H.K.); johnsperfume@gmail.com (J.H.J.); 2College of Korean Medicine, Kyung Hee University, 26 Kyungheedae-ro, Dongdaemun-gu, Seoul 02447, Korea; 3Department of New Biology, Daegu Gyeongbuk Institute of Science and Technology, Daegu 42988, Korea

**Keywords:** wound healing, ferulic acid, shilajit, K6α, β-catenin, Nrf2, keratinocytes

## Abstract

Injured tissue triggers complex interactions through biological process associated with keratins. Rapid recovery is most important for protection against secondary infection and inflammatory pain. For rapid wound healing with minimal pain and side effects, shilajit has been used as an ayurvedic medicine. However, the mechanisms of rapid wound closure are unknown. Here, we found that shilajit induced wound closure in an acute wound model and induced migration in skin explant cultures through evaluation of transcriptomics via microarray testing. In addition, ferulic acid (FA), as a bioactive compound, induced migration via modulation of keratin 6α (K6α) and inhibition of β-catenin in primary keratinocytes of skin explant culture and injured full-thickness skin, because accumulation of β-catenin into the nucleus acts as a negative regulator and disturbs migration in human epidermal keratinocytes. Furthermore, FA alleviated wound-induced inflammation via activation of nuclear factor erythroid-2-related factor 2 (Nrf2) at the wound edge. These findings show that FA is a novel therapeutic agent for wound healing that acts via inhibition of β-catenin in keratinocytes and by activation of Nrf2 in wound-induced inflammation.

## 1. Introduction

Wound healing is a dynamic biological process with complex, diverse interactions at the molecular level that are only partially understood. Many studies in recent decades have contributed to a better understanding of the mechanisms of the wound repair process and the causes and results of delayed wound healing. In addition, acute wounds are still frequently reported to cause health trouble, with 11 million people suffering from wounds and approximately 300,000 patients hospitalized annually in the United States. [1]. Typically, wound healing is a well-organized process that lead to predictable tissue repair in which platelets, keratinocytes, immune surveillance cells, microvascular cells, and fibroblasts play key roles in the restoration of tissue integrity [2]. Under the conditions of injury, keratinocytes induce keratin 6 (K6) isoforms, keratin 16 (K16), and keratin 17 (K17). The expression of K6, K16, and K17 persists as wound-activated keratinocytes migrate to the site of injury [3]. This transcriptional event of keratins occurs at the expense of keratin 1 and keratin 10, and correlates with marked modulation of the morphology and other properties of keratinocytes [3,4]. The induction of K6 isoforms and K16 has functional effects on the epithelialization at the wound edge; they are essential to the homeostasis of keratin integrity in the epidermis [5]. To assist recovery from acute wounds, the development of safer therapeutic methods has been studied. The β-catenin signaling pathway has been suggested as a good candidate for wound therapy, as β-catenin is involved in the regulation of the wound size in mesenchymal cells. The activation of β-catenin is known to contribute to chronic wounds in keratinocytes [6,7]. Furthermore, β-catenin is increased during the proliferative phase of cutaneous wound healing [8,9,10] and regulates the proliferation, motility, and invasiveness of mesenchymal fibroblast-like cells under wound conditions [11].

The Wnt signaling pathway plays important roles in many aspects of the control of cell proliferation, survival, adhesion, and migration through both canonical β-catenin-dependent and non-canonical β-catenin-independent signaling [12,13]. During the wound process, interestingly, epidermal keratinocytes and fibroblasts have opposite molecular patterns under canonical β-catenin-dependent signaling conditions. For example, nucler presence of β-catenin and elevated c-myc expression showed impaired migration of keratinocytes in the human epidermis in chronic wound conditions [7]. On the other hand, valproic acid (VA), an inhibitor of glycogen synthase kinase-3 beta (GSK3β), has demonstrated its efficacy for wound repair in dermal fibroblasts [14], as well as enhancement of hair growth [15] and the induction of hair regeneration through the activation of β-catenin through the canonical β-catenin-dependent signaling [16,17]. Given the crucial role of β-catenin signaling in normal homeostasis and repair after injury, the regulation of β-catenin as an appropriate homeostatic response may be a useful therapeutic target for the enhancement of wound healing. However, VA therapy may cause critical side effects, such as hepatotoxicity, pancreatitis, and hyper-ammonemia [18,19].

In a previous study, we screened natural resources to develop therapeutic agents for wound healing and identified biological indicators of proliferation and migration, such as the gene regulation and signaling pathways in wound healing by using microarray [20]. In the present study, to develop a safe therapeutic target for wound healing with minimal side effects, we tested the wound healing effects of shilajit (SH) in ayurvedic medicine under in vivo and ex vivo conditions and analyzed the biological signaling pathway after treatment with SH using microarray testing. SH is a blackish-brown exudate, which forms over centuries through the progressive decomposition of natural substituents by microorganisms, and is found in the Himalayans between India and Nepal [21]. In Ayurveda, the traditional medicine system of India, SH has been prescribed for centuries to patients with several diseases, including edema, tumors, epilepsy, insanity, diabetes, and skin wounds [22]. As it has been used safely and efficiently for such a long time that SH has attracted the attention of researchers for further studies. For example, researchers attempted to identify the active ingredient in SH, and found that SH contained several natural components, such as humic acid, fulvic acid, tannic acid, gallic acid, and ferulic acid (FA) [23,24,25]. Among the various components, studies have suggested anti-cancer effects of FA in cancer cell lines [26]. The wound healing efficacy of FA has been tested in a rat model of diabetes [27]. However, most of the previous studies did not investigate the mechanism of action of SH. In this study, to find the active compound, we performed liquid chromatography–electrospray ionization–mass spectrometry (LC-ESI-MS) on SH samples. Finally, we excavated the active compound, ferulic acid, from SH and suggested that FA promoted the re-epithelialization of wound conditions in vivo and inhibited β-catenin in primary keratinocytes through the enhancement of migration ex vivo. In the histological analysis of a full-thickness wound edge, FA inhibited translocation of β-catenin in epidermal keratinocytes. Furthermore, FA alleviated wound-induced inflammation via activation of nuclear factor erythroid-2-related factor 2 (Nrf2), which is associated with cellular protection at the wound edge. These results implied that FA induced migration by inhibiting nuclear accumulation of β-catenin and induced rapid repair by activating Nrf2.

## 2. Materials and Methods

### 2.1. Shilajit Preparation and Ferulic Acid

Purified shilajit (PRIMAVIE^®^) was purchased from Natreon Inc. (New Brunswick, NJ, USA), India. Shilajit was collected from 2400 m altitude in the rocky mountain of Beshtor, Tashkent, Uzbekistan. A sample specimen is kept in the laboratory with the specific code number. For qualitative analysis using LC-ESI/MS, shilajit was extracted by water, acetonitrile, or methanol of 100% HPLC grade for 72 h at 45 °C in sonicated conditions (Branson, MO, USA). Ferulic acid was purchased from Sigma-Aldrich (Saint Louis, MO, USA).

### 2.2. Cell Cultures

Cell cultures of primary keratinocytes were harvested and purified as previously described [5]. Neonatal mouse skin was floated onto 0.25% trypsin and incubated overnight. Cells were scraped into CnT-57 media (CELLnTEC, Bern, Switzerland) and live cells were purified in Lymphoprep solution (AXIS-SHIELD, Dundee, UK) and cultured in CnT-57 media. ICR keratinocytes were harvested from newborn mice and seeded on six-well plates (TPP; Sigma-Aldrich,) for primary culture in CnT-57 media. Primary fibroblasts were purified in Lymphoprep solution (AXIS-SHIELD) and were cultured by selecting supernatants of neonatal mouse skin in DMEM media of high glucose (Lonza, Walkersville, MD, USA) containing 10% heat-inactivated fetal bovine serum (FBS; Lonza), penicillin, and streptomycin (Thermo Fisher Scientific, Vantaa, Finland). Cell cultures were incubated at 37 °C in humidified air and 5% CO_2_ atmosphere [28]. The 308 cell line, which is a mouse keratinocyte cell line, was received from Pierre A. Coulombe (Bloomberg School of Public Health, Johns Hopkins University, Baltimore, MD, USA) and was described previously [29]. The 308 cell line was cultured in DMEM media of high glucose (Lonza) containing 10% heat-inactivated fetal bovine serum (Lonza), penicillin, and streptomycin. Cell cultures were incubated at 37 °C in humidified air and 5% CO_2_ atmosphere [30].

### 2.3. Real-Time Cell Analysis (RTCA) Measurement

Growth curves were performed using the xCELLigence System (Roche, Basel, Switzerland) in 96-well plates. Cell Index was monitored every fifteen minutes throughout the experiment. For proliferation assays, cells were seeded in triplicate in 18-well culture plates and grown for 72 h after treatment of shilajit or ferulic acid at 37 °C in humidified air and 5% CO_2_ atmosphere.

### 2.4. Real-Time Quantitative PCR

Total RNA samples from primary keratinocytes, derived from mouse back skin, were isolated. Following this, the cDNA was hybridized from 1 μg of the total RNA with a LeGene first strand cDNA synthesis system (LeGene Bioscience, San Diego, CA, USA) [31]. Several expression levels of mice keratin family (*Krt6a, Krt5*, and *Krt14*) were determined with a quantitative PCR test, as described in the manufacturer’s protocol (Applied Biosystems, Foster City, CA, USA). The 2^−ΔΔCt^ value compared to the normal mice sample was determined with StepOne software (Applied Biosystems). Gapdh was used as a house keeping gene and endogenous control. The sequence of the forward and reverse primer was 5′-CATGGCCTTCCGTGTTCCTA-3′ and 5′-GCGGCACGTCAGATCCA-3′ for the Gapdh gene, 5′-CCCTCTGAACCTGCAAATCG-3′ and 5′-GATCTGCTCCCTCTCCTCAGT-3′ for the Krt6a, 5′-TGCCCTGCCGTTTCTCTACT-3′ and 5′-TGATCTGCTCCCTCTCCTCA-3′ for the Krt5, and 5′-ACGAGAAGATGGCGGAGAAG-3′ and 5′-CTCTGTCTTGCTGAAGAACCATTC-3′ for the Krt14.

### 2.5. Western Blotting

After mouse primary keratinocytes and skin tissues were treated with shilajit or ferulic acid, the cells and tissues were lysed and the total protein concentrations were determined by Bradford reagent (Bio-Rad, Hercules, CA, USA). Equal amounts of lysates resolved using sodium dodecyl–polyacrylamide gel electrophoresis (SDS-PAGE) were transferred to a nitrocellulose membrane, then the membrane was blocked with 1 x TBS containing 0.1% Tween 20 and 5% skim milk or 2% BSA for 1 h at room temperature [32]. After the blocking, the blots were then incubated with specific primary antibodies for β-catenin (BD Bioscience, San Jose, CA, USA), K6α (Covance, Cranford, NJ, USA), or β-actin (Santa Cruz Biotechnology, Dallas, TX, USA) overnight at 4 °C. Blots were washed with TBST and incubated HRP-conjugated anti-mouse or anti rabbit IgG secondary antibodies (Santa Cruz Biotechnology) for 1 h at room temperature. After three washes, the membranes were detected using an enhanced chemiluminescence (ECL) kit (Millipore, Burlinton, MA, USA) [33].

### 2.6. Animals

Commercially available ICR mice purchased from DBL (Eumseong-gun, South Korea), were used for the experiments and adapted under constant conditions. The ICR littermates (0-1 day old mice) were used for skin explant culture. ICR mice (8 weeks old mice) were used for acute wound models. All studies were performed at Kyung Hee University. All animal study protocols were approved by the Institutional Animal Care and Use Committee (IACUC) of Kyung Hee University. The approval number for the animal experiment was KHUASP(SE)-14-046.

### 2.7. Wound Closure Test

Full-thickness excisional wounds were developed in the middle of the back skin with a 4 mm disposable biopsy punch (Kai industries, Seki, Japan) in ICR mice (8 weeks old mice). Five groups were prepared and used in the study. The control (Con) group was untreated, while the VA group was treated with only Vaseline petroleum jelly (VI-JON LABORATORIES, INC, Saint Louis, MO, USA). SH 1% and SH 10% groups received varying concentrations of the shilajit (shilajit 0.1 g and 1 g per 10 g of the Vaseline base, respectively). The FA 1% group received 1% concentration of the FA (FA 0.1 g per 10 g of the Vaseline base). In this study, Vaseline was used as the vehicle. All mice were observed for 7 days, and at 1, 3, 5, and 7 days wound images were taken with a camera (Nicon, Tokyo, Japan) and the wound closures were recorded using ImageJ software.

### 2.8. Skin Explant Culture

Ex vivo explant culture of 1 d old mouse skin was performed as described previously [34]. Using 4 mm punches (Kai industries, Seki, Japan), circular skin biopsies were obtained and plated with medium in 24-well dishes (SPL, Pochen-si, South Korea). A subset of explants was incubated in 24-well plates (SPL) and treated with shilajit (1 μg/mL) after 48 h. After treatment, all explants were incubated for 24 d to confirm the cellular outgrowths and mRNA levels using microarrays. Outgrowth of epithelial cells was confirmed by hematoxylin and eosin staining (H&E). The epithelial cells were fixed in 4% paraformaldehyde for 20 min. After fixation, the cells were stained with H&E solution [35]. The outgrowth of H&E-stained cells was scanned by microscope (Olympus, Tokyo, Japan) and measured using ImageJ freeware at 12 d and 24 d, respectively.

### 2.9. Immunofluorescence

The isolated cells were fixed in 4% paraformaldehyde for 20 min and then incubated with 3% H_2_O_2_ in 0.1M PBS for 30 min to remove endogenous peroxidase activity. The cells were blocked for 2 h at room temperature with a solution containing 5% normal goat serum, 2% bovine serum albumin, 2% fetal bovine serum, and 0.1% triton X-100 in PBS. The primary keratinocytes derived from skin explants (P0-P1) were first incubated overnight with rabbit anti-keratin 6α (1:200, Covance) and then incubated with Alexa Fluor 488 conjugate donkey anti-rabbit IgG secondary antibody (1:000; Invitrogen, Carlsbad, CA, USA) for 1 h at room temperature. The primary keratinocytes and fibroblasts derived from back skin (P0-P1), were incubated overnight with mouse anti-c-Myc (1:100; Santa cruz) and with rabbit anti-beta-catenin (1:200; Cell signaling technology, Danvers, MA, USA), then incubated with Alexa Fluor 647 conjugate goat anti-mouse IgG secondary antibody and with Alexa Fluor 488 conjugate donkey anti-rabbit IgG secondary antibody for 1 h at room temperature. DAPI (Invitrogen) and Alexa Fluor 594 were visualized using a FluoView FV1000 confocal microscope (Olympus, Tokyo, Japan). Isolated wounded tissues were fixed in 4% paraformaldehyde, embedded in paraffin wax within 24 h of removal, then deparaffinized in xylene and ethanol. The duodenal sections were incubated with 3% H_2_O_2_ in 0.1M PBS for 30 min to remove endogenous peroxidase activity. The sections were blocked for 2 h at room temperature with a solution containing 5% normal goat serum, 2% bovine serum albumin, 2% fetal bovine serum, and 0.1% triton X-100 in PBS. The skin sections of wounded edges were incubated overnight with mouse anti-beta-catenin (1:200; BD Bioscience) and rabbit anti-keratin 6β, then incubated with Alexa Fluor 647 conjugate goat anti-mouse IgG secondary antibody (1:1000; Abcam, Cambridge, UK) and with Alexa Fluor 488 conjugate donkey anti-rabbit IgG secondary antibody (1:1000; Abcam, Cambridge, UK) for 1 h at room temperature. The immunofluorescence intensity values were measured using a FluoView FV1000 confocal microscope and analyzed using Mann–Whitney U tests in GraphPad Prism 5 software.

### 2.10. Microarray

Biotinylated cRNA samples were prepared according to the standard Affymetrix protocol from 500 ng total RNA (Expression Analysis Technical Manual, 2001, Affymetrix). Following fragmentation, 15 mg of aRNA was hybridized for 16 h at 45 °C on a GeneChip Mouse Genome Array. GeneChips were washed and stained in the Affymetrix Fluidics Station 450. GeneChips were scanned using the Affymetrix GeneChip 3000 7G scanner. The data were analyzed with RMA using Affymetrix default analysis settings with global scaling as the normalization method. The trimmed mean target intensity of each array was arbitrarily set to 100. The normalized and log-transformed intensity values were then analyzed using GeneSpring GX 12.6 (Agilent Technologies, Santa Clara, CA, USA). Fold change filters included the requirement that the genes be present in at least 200% of controls for upregulated genes and lower than 50% of controls for downregulated genes. Hierarchical clustering data were clustered groups that behaved similarly across experiments using GeneSpring GX 12.6 (Agilent Technologies). The clustering algorithm involved Euclidean distance and average linkage.

### 2.11. LC-ESI-MS

Chromatographic separation of shilajit using an Agilent 1290 Infinity LC (Agilent Technologies, Waldbronn, Germany) instrument was performed using a Zorbax Eclipse Plus C18 column (50 mm× 2.1 mm i.d., 1.8 μm, Agilent) that employed a mobile phase composed of water (A) and acetonitrile (B), each containing 0.1% formic acid (Sigma Aldrich). The gradient program was: 0–5 min (5% B in A), 5–15min (5–90% B), 15–20 min (90% B), and 20–21min (90–5% B). The column was then equilibrated with 5% B for 5 min at a flow rate of 300 μL/min. A sample of 2 μL was injected into the column using a thermostated HiP-ALS autosampler. The HPLC system was interfaced to the MS system using an Agilent 6550Accurate-Mass Q-TOF instrument (Agilent Technologies) equipped with a Jet Stream ESI source operating in negative ion mode. Mass spectra were acquired at a scan rate of 1.0 spectra/s, with a mass range of 100–1700 *m*/*z*.

### 2.12. Ethics Statement

All animal study protocols were approved by the Institutional Animal Care and Use Committee (IACUC) of Kyung Hee University. The approval number for the animal experiment was KHUASP(SE)-14-046 (13 April 2015). The 308 cell line, which is a mouse keratinocyte cell line, was received from Pierre A. Coulombe (Bloomberg School of Public Health, Johns Hopkins University, Baltimore, MD, USA).

### 2.13. Statistical Analysis

Data are expressed as the means ± S.E.M. Graph Pad Prism 5 (Graph Pad software, San Diego, CA, USA.) was used for the statistical analysis. The statistical significance of each bar chart was measured using one-way ANOVA with Dunnett’s post-hoc (for in vitro studies) or Tukey’s post-hoc (for in vivo studies) test. For in vivo studies, two-way ANOVA with Tukey’s post-hoc (to compare more than 3 groups) or Bonferroni’s post-hoc (to compare 2 groups) test was performed. Here, *p* < 0.05 was considered significant.

## 3. Results

### 3.1. SH Induces Keratinocyte Migration and Its Active Component Is FA

Although SH is used for wound tissue healing in ayurvedic medicine, the mechanism of action of SH and the active components are still unknown. To discover the biological action of SH under wound conditions, we induced an acute wound model in mice and observed wound closure with or without SH. First, we induced an epidermal wound by using a 4 mm skin punch (Experimental process, Appendix A) and observed wound closure in the presence or absence of SH (Figure 1a). In the treatment group with SH 10%, we observed more significant wound closure than the basal amount after day 3 (Figure 1b). The number of days needed to close 50% of the wound area was as follow: basal (BSL, 5.43 ± 0.89 day), vehicle (Ve, 5.49 ± 0.25 day), SH 1% (4.08 ± 0.107 day), and SH 10% (3.16 ± 0.09 day) (Figure 1b). To validate wound conditions for both non-wound and wound groups (basal group) on day 7, skin full-thickness was evaluated by wound-induced keratin (K6α wound marker) using immunofluorescence staining (Appendix A).

When wound edges healed, migration of keratinocytes contributed to rapid wound closure. Mazzalupo et al. reported that the primary keratinocytes derived from the back skin of mouse pups were used to evaluate quantitative outgrowth [34]. To further evaluate wound-healing-associated migration of keratinocytes, we prepared a primary culture derived from the back skin of explant culture in pups born within day 1 (24 h) and observed the outgrowth of epithelial keratinocytes from the back skin of explant culture on day 12 or 24 (Figure 1c). Through H&E staining, we easily distinguished the outgrowth of primary keratinocytes, measured the migrated area of primary keratinocytes, and evaluated the quantitative migration effects with or without SH. After treatment with SH (1 μg/mL), outgrowth of primary keratinocytes was significantly enhanced from day 12 to day 24. The fold changes in outgrowth of keratinocytes were 4.94 ± 0.56 at day 12 and 5.13 ± 0.58 at day 24 (Figure 1d). A schematic figure of the ex vivo skin explant culture is represented in Appendix A.

To cover SH-mediated migration associated with biological molecules, we performed transcriptomic analysis and a microarray of primary keratinocytes in the growth area of skin explant cultures with or without SH. In the microarray, we arranged functional categorization. The alteration of gene expression of at least two-fold was sorted into 17 categories: apoptosis, behavior, cell adhesion, cell cycle, cell differentiation, cell migration, cell proliferation, growth, homeostasis, immune response, inflammatory response, lipid metabolism, response to stress, signal transduction, transcription, and transport (Figure 1f,g). We deduced that the major alterations in mRNA expression were the upregulation of both keratingenes (Table 1), and junctional complex genes (Table 2) but, the downregulation of Ctnnb1 (Table 3). Next, to evaluate the preliminary mRNA expression results from the microarray analysis, we analyzed the mRNA level of Krt6a related to wound-induced keratin by using qPCR. Krt6a was increased by more than two-fold in the SH-treated group on day 12 (2.38 ± 0.17) and day 24 (2.22 ± 0.24) (Figure 1e). In addition, the mRNA expression levels of Krt6a, Krt5, and Krt14, and the protein expression of K6α were increased by SH treatment of primary keratinocytes (Appendix A). Under wound edges after day 7, SH treatment showed effective re-epithelialization on epidermis (Appendix A). The SH-mediated migration suggested that an interaction between increased Krt6a and Ctnnb1 may be associated with rapid wound healing.

After determining the biological evidence for SH-mediated migration associated with modulation of both Krt6a and Ctnnb1, we investigated the active compound involved in rapid wound healing. Shalini et al. reported that SH contained natural components such as tannic acid, gallic acid, caffeic acid, and ferulic acid (FA) [25]. Although a previous study reported that FA had healing effects for diabetic injuries via amelioration of hyperglycemia, our results focused on the biological and pharmacological effects of a topical skin ointment treatment during wound healing [27]. Here, we accessed weather FA as a topical medication induced wound healing or not via general blood glucoses. Before testing the FA ointment, we analyzed the SH components. Shalini et al. showed that SH contained 37.55 μg/g of ferulic acid in their study and the ratio between SH and FA was 1:2.66 × 10^−8^ (shilajit: Ferulic acid). In our sample, we analyzed which SH components were involved using total ion chromatography (Appendix A), and the ratio between SH and FA was 1:1.211 × 10^−5^ (Appendix A). FA was detected among the natural components of SH (Appendix A).

Subsequently, to test the wound healing effects associated with wound closure and migration, we firstly observed wound closure treatment with FA 1% in an acute wound model (Figure 1h). In treatment with FA 1%, significant wound closure was observed on day 3 (Figure 1i). The numbers of days required for the closure of 50% of the wound area were as follows: BSL (5.92 ± 0.27 days), Ve (5.74 ± 0.26 days), and FA (3.99 ± 0.19 days) (Figure 1i). Next, we tested the migration of primary keratinocytes and their morphology in ex vivo cultures in the presence or absence of 100 nM FA (Figure 1j). We observed that FA treatment had three biological features in common with SH treatment. FA enhanced K6α expression (Figure 1j) and induced outgrowth of keratinocytes, similar to SH treatment on day 24 (Figure 1k). The change in outgrowth in the presence of FA was 3.48 ± 0.66 on day 12 (Figure 1k). FA decreased Ctbbn1 expression (Figure 1j). The fold change of the Ctbbn1 expression was 0.4 ± 0.02 (Figure 1l). In the histological study after treatment of SH or FA, FA treatment was the most effective wound recovery under wound edge at day 7d (Appendix A). Moreover, qPCR analysis was performed for *Krt6a, Krt6b, Krt16*, and *Nqo1* (Appendix A). These results showed that FA derived from SH may induce wound closure and migration by modulation of Krt6 and Ctbbn1.

### 3.2. SH and FA Induce Rapid Homeostatic Response of β-Catenin in Dermal Fibroblasts but Not in Epidermal Keratinocytes

Microarray data for cells treated with SH showed the upregulation of several keratins (Table 1) and molecules of the extracellular matrix (Table 2). However, the mRNA level of β-catenin was most downregulated in the epithelial keratinocytes of the skin explant culture (Figure 1j and Table 3). To further examine the protein expression of the genes with altered mRNA levels (i.e., the decreased Ctbbn1 and the increased Krt6a) in epidermal keratinocytes of the skin explant culture, histology study using immunofluorescence staining of K6α and β-catenin was performed on wound edges. In accordance with the wound closure results after treatment with SH and FA (Figure 1b,g), we observed that the significant time points of wound healing were day 3 and day 7. Then, we performed a histological study of wound edges on day 3 and day 7 after treatment with SH and FA. We measured the protein expression of β-catenin and K6α in full-thickness wound edges on days 3 and 7 after treatment with SH and FA (Figure 2). The localization of β-catenin on the wound edge of the dermis was remarkably increased by treatment with SH 10% (Figure 2a). Dose-dependent increased β-catenin expression on the wounded dermis was evaluated using Western blotting on day 3. The fold change of β-catenin expression against non-treatment was 1.19 for SH 1% and 1.4 for SH 10% (Figure 2c). The localization of K6α, the wound-induced marker, appeared as a thick line on the wound edge of the epidermis after treatment with SH 10% (Figure 2a). K6α expression on the wounded epidermis was dose-dependently increased, as shown by western blotting on day 3. The fold change for increased expression of K6α against non-treatment was 1.11 for SH 1% and 1.44 for SH 10% (Figure 2c). After post-wounding day 7, interestingly, the localization of β-catenin on the wound edge of the dermis with SH 10% seemed to be more decreased than basal group (Figure 2b). This decreasing tendency of β-catenin expression was evaluated using Western blotting. The fold change for decreasing expression of β-catenin was 0.4 for SH 10% compared with basal rates (Figure 2d). However, the localization of K6α continue to increase in epidermal keratinocytes of the wound edges (Figure 2b). The state of increase of K6α was evaluated using Western blotting. The fold change for increasing expression of K6α was 2.2 for SH 10% compared with basal rates (Figure 2d).

To further investigate whether FA induced wound healing to modulate K6α and β-catenin in common with SH, we tested the localization and quantitative expression of K6α and β-catenin under wound edges on days 3 and 7 (Figure 2e–h). Similar to the modulation by treatment with SH 10%, FA increased the localization and expression of β-catenin on day 3 (Figure 2e,g) and subsequently decreased the localization and expression of β-catenin on day 7 in dermal fibroblasts (Figure 2f,h). In addition, FA treatment resulted in greater accumulation of β-catenin than basal conditions on day 7 in the epidermal keratinocytes on the wound edge (Figure 2f). During wound healing, additionally, FA treatment continued to increase the localization and expression of K6α on day 3 (Figure 2e,g). These results suggested that FA exerted similar wound healing effects to SH, modulating the downregulation of β-catenin and the upregulation of K6α, implying that FA modulated the expression of β-catenin and K6α at the wound edge and enhanced wound healing through the promotion of keratinocyte migration. These results suggested that SH enhanced wound closure by modulating downregulation of β-catenin on the dermis and upregulation of K6α on the epidermis. Additionally, FA was an active wound healing agent for modulation of both β-catenin and K6α under the wound edge.

### 3.3. FA Suppresses Nuclear Accumulation of β-Catenin in Epidermal Keratinocytes

As the nuclear accumulation of β-catenin caused the induction of c-myc, it contributed to the suppression of migration and the maintenance of epidermal chronic wounds in human epidermal keratinocytes [7]. Here, we hypothesized that if FA inhibited translocation of nuclear β-catenin or disturbed the accumulation of nuclear β-catenin, the delayed migration of keratinocytes might be improved. To evaluate whether FA inhibited the nuclear accumulation of β-catenin, we observed this phenomenon in primary keratinocytes treated with and without FA (Figure 3a). FA treatment significantly inhibited the nuclear translocation of β-catenin (Figure 3b) and decreased the nuclear translocation of c-myc, a downstream molecule of β-catenin (Figure 3c). To further examine the effects of FA on the inhibition of the nuclear accumulation of β-catenin, we induced the overexpression of β-catenin by using LiCl and observed the protein localization of β-catenin and c-myc (Figure 3a, white triangle). The LiCl-induced nuclear accumulation of β-catenin delayed keratinocyte migration and wound healing; if this was inhibited by FA, we would expect this to contribute to keratinocyte migration in wound healing. The IF analysis revealed that FA restricted the nuclear accumulation of β-catenin and c-myc in primary keratinocytes treated with LiCl (Figure 3a–c). In addition, FA blocked intracellular β-catenin and c-myc expression in the keratinocytes (Figure 3d–f). Our results showed that 10% SH and FA increased localization and expression of β-catenin in dermal fibroblasts on the wound edge on day 3 (Figure 2a,e). To examine the differential effects of FA, we additionally observed localization and expression of both β-catenin and c-myc in primary fibroblasts (Figure 3g,j). FA increased translocated β-catenin and c-myc expression in primary fibroblasts, as shown by IF analysis (Figure 3g–i). Subsequently, there were no changes in β-catenin and c-myc expression in fibroblasts after LiCl treatment (Figure 3j–l). These results implied that FA’s actions of migration and wound closure resulted from the inhibition of β-catenin accumulation on epidermal keratinocytes and the translocation of β-catenin into the nucleus of dermal fibroblasts.

### 3.4. FA Ameliorated Wound-Induced Inflammation by Activation of NRF2

After tissue injury, the inflammatory response plays important roles in wound healing. To evaluate whether FA had anti-inflammatory or protective effects in FA-mediated rapid wound healing, we observed major inflammatory markers, such as COX-2 and iNOS, as well as the protective marker Nrf2. In the wound edge, only iNOS localization was significantly decreased on day 7 of FA treatment (Figure 4a). Regarding Nrf2 localization, however, FA treatment resulted in effective enhancement on both day 3 and day 7 (Figure 4b). To further evaluate the FA-mediated iNOS decrease, we tested COX-2 and iNOS expression in RAW 264.7 under LPS-mediated inflammation conditions. After 100 μM of FA treatment, only iNOS expression was significantly decreased (Figure 4c). A previous study suggested that FA had anti-inflammatory effects via inhibition of NF-κB and activation of Nrf2 [36]. However, the molecular interaction between iNOS and Nrf2 related to inflammatory signaling was still unknwon. To investigate the mode-of-action of FA in terms of anti-inflammatory effects, we performed systemic knockdown using siRNA from Nfe2l2 and observed COX-2 and iNOS expression. Under Nfe2l2 knockdown conditions, the FA-mediated iNOS decrease disappeared in the RAW 264.7 mouse macrophage (Figure 4d). The evaluation of the knockdown of Nfe2l2 is shown in Figure 4e. These results suggested that FA alleviated wound-induced inflammation by decreasing iNOS expression, and that this iNOS decrease was mediated by FA-mediated Nrf2 activation.

### 3.5. FA Induces Proliferation in Keratinocytes

During the regeneration of the epidermis after wounding, the proliferation of keratinocytes plays a crucial role. To determine the cellular proliferative effects of wound healing by SH and FA, we performed real-time cell analysis (RTCA). RTCA was automatically recorded by using the xCELLigence System in real time. The Kera 308 cells, a mouse keratinocyte cell line [29], were seeded at 5 × 103 cells in each E-plate. The dose curves were obtained between 1 μg/mL to 1000 μg/mL SH (Figure 5a) and between 1 nM and 1000 nM FA for 72 h (Figure 5b). The SH treatment group values were more increased than the basal values. The values for the area under the curve are as follows: BSL (1.02 ± 0.02), 1 (1.34 ± 0.03), 10 (1.3 ± 0.04), 100 (1.33 ± 0.03), and 1000 (1.29 ± 0.01) (Figure 5a). The FA treatment values from 1 to 100 nM were increased more than the basal values. The values for the area under the curve were as follows: BSL (1 ± 0.01), 1 (1.17 ± 0.05), 10 (1.25 ± 0.04), 100 (1.4 ± 0.03), and 1000 (1.01 ± 0.02) (Figure 5b). To further evaluate the proliferative effects of SH and FA at the wound edge, we observed the proliferation marker, Ki67, in the wound edge after treatment with SH and FA. SH and FA induced a significant increase in Ki67-positive cells in the wound epidermis 7 days after wounding. The values for Ki67-positive cells were as follow: BSL (8.6 ± 1.56), SH 10% (20.2 ± 1.56), and FA (43.8 ± 3.42) (Figure 5c). The proliferative enhancement induced by FA was more than two-fold greater than that of SH. In addition, genes associated with proliferation, such as integrin-linked kinase, chemokine ligands, and fibroblast growth factor, were altered by SH treatment (Table 3). These results suggested that keratinocytes were enhanced to proliferation in response to FA treatment in common with SH. In addition, FA treatment affected the dose-dependent increase. These results suggested that FA might be induced in wound healing as an active compound by enhancement of proliferation and Ki67-positive cells.

## 4. Discussion and Conclusions

Under wounding conditions, the epithelial sheets of metazoans immediately trigger the repair mechanism to restore the condition and function of epithelial cells. The epidermis acts as a protective barrier; the degree of this action is dependent upon the dynamic degree of integrity of the keratin network [37,38,39,40]. Simultaneously, to repair the epidermal wound edge, epithelial stem cells from the hair follicle bulge migrate into the epidermis during wound healing [41].

In the present study, we demonstrated that SH and FA treatment induced rapid wound closure in acute wounds and the migration of epidermal keratinocytes in ex vivo culture. SH and FA affected the regulation of keratin genes and β-catenin (Figure 1). The induction of K6α under wounding conditions plays an important structural role in the wound epithelialization of keratinocytes and the maintenance of keratinocyte integrity [5]. K6α regulates homeostasis during recovery through a direct interaction with Src; thus, the interaction between K6α and Src may be a trigger to dampen migration in the wound edge [4,42]. Paradoxically, the loss of the K6 isoform of keratinocytes resulted in enhanced epithelial migration in the wound edge [43]. Recently, the K6 isoform modulated keratinocyte migration by regulation of cell–matrix and cell–cell adhesion dynamics, which promoted essential collective cell migration and wound healing [44]. In further studies of the development of therapeutic agents in wound healing, evaluation of migration, the modulation of the K6 isoform, and its interaction proteins associated with cytoarchitecture, adhesion, and force generation should be considered at the wound edge [45,46].

Second, our results indicated that SH and FA treatment increased K6α and β-catenin expression in the acute wound on day 3. However, on day 7, the increase in K6α was maintained but the increase in β-catenin was dissipated by treatment with SH and FA. SH and FA appeared to regulate protein expression in epidermal keratinocytes and dermal fibroblasts (Figure 2). In mechanical stress conditions, such as at the wound edge, β-catenin plays an essential role in the epidermal barrier function. Therefore, the loss of β-catenin in the embryonic epidermis causes neonatal lethality through the activation of K6 [47]. Wnt/β-catenin signaling is a functional network for the regulation of keratins and is generally involved in areas such as stem cells and developmental biology [48]. When the Wnt pathway is inactivated, the key mediator of Wnt signaling, β-catenin, participates in the adherens junction, where it contributes to the stabilization of cell-to-cell interactions [49]. When the Wnt pathway is activated, β-catenin functions not only in the cytoplasm, where β-catenin levels are tightly controlled by processes regulating protein stability, but also in the nucleus, where β-catenin is involved in transcriptional regulation and chromatin interactions [50].

Third, treatment with FA, an active compound involved in wound healing, resulted in disturbed nuclear β-catenin expression after LiCl treatment in primary keratinocytes, whereas nuclear accumulation of β-catenin was activated by FA treatment in primary fibroblasts (Figure 4). In hair follicle progenitor cells, β-catenin signaling plays an important role in the active growth phase of the hair cycle [51] and contributes to proliferation, survival, and epidermal homeostasis [52]. In the continuous nucleus accumulation of β-catenin, however, keratinocyte migration and wound healing are suppressed by the induction of the c-myc activation in chronic wounds in humans [7]. The deregulation of c-myc depletes epidermal stem cells, resulting in an inability to restore wound sites and reduce β1 integrin expression, which is involved in keratinocyte migration [53]. Thus, FA treatment suppressed nucleus accumulation of β-catenin and c-myc in primary keratinocytes, indicating wound healing via increasing migration.

Furthermore, the loss of β-catenin caused accelerated wound healing owing to reduced numbers of fibroblasts and the failure of differentiation into follicular keratinocytes [54,55]. However, in cutaneous wounds, the activation of the Wnt/β-catenin pathway induces cutaneous wound healing in dermal fibroblasts through the inhibition of GSK3β by valproic acid [16], because GSK3β is involved in fibrosis through β-catenin signaling in dermal fibroblasts [56]. In addition, CXXC5, a negative feedback regulator of the Wnt/β-catenin pathway, enhanced collagen production in cutaneous wound healing [57]. However, evidence of the therapeutic effect is limited in clinical trials of cutaneous wound healing. The problem of delivery is more tractable, as small-molecule drugs (typically <550 Da) pass through the dermal layer [58]. The side effects of small-molecule drugs include low specificity and accumulation in specific organs, which may result in severe toxic effects; for example, valproic acid affects chemicals in the body and is involved in seizures and epilepsy [59]. Although the previous study suggested that the regulation of β-catenin showed opposing mechanisms between keratinocytes and fibroblasts in wound healing [55], the development of therapeutic agents targeting both keratinocytes and fibroblasts has been poorly studied. Our results suggest that FA modulates the gene expression of β-catenin through the differential regulation of keratinocytes and fibroblasts and leads to epidermal migration and proliferation as a safe therapeutic target for wound healing. In a recent study, the wave complex, which is involved in structural organization and plays a mechanical function in cells, was essential in epidermal development by suppressing Wnt signaling [60]. In wound treatment, the control of Wnt signaling in epidermal keratinocytes may lead to well-established re-epithelization with minimal epidermal trauma. In the present study, we originally reported that FA modulated β-catenin in primary keratinocytes and fibroblasts. Our findings extend the current understanding of the pathogenesis of keratin disorders, as well as the knowledge of rapid wound healing. For example, the genetic mutation of K6a (~50% of cases), K6b, K16, or K17 causes pachyonychia congenita (PC), which is a keratinizing disorder and is a phenotypic feature of focal plantar keratoderma [61]. To treat hyperkeratotic disorders such as PC, retinoids or statins are therapeutic options. [62,63]. However, retinoids are not specific for a given keratin or keratin-related gene, and retinoids and statins may cause side effects through the inhibition of epidermal differentiation and hepatotoxicity [64,65,66]. In particular, statins decrease K6a expression by downregulation of K6a promoter activity [63], and may promote neurological injury of the hippocampus and spinal cord via enhancement of the Wnt/β-catenin signaling pathway [67,68]. Further studies to understand interaction of keratins and Wnt/β-catenin signaling will provide treatment strategies for PC and other keratin-related disorders. Indeed, FA could be a plausible treatment option, as FA has been shown to induce compensatory keratin gene expression to overcome the effects of the mutated keratin gene and to regulate differential signaling modulation for fibroblasts and keratinocytes. Sulforaphane (SF), which is chemopreventive agent extracted from broccoli sprouts [69], regulates the induction of K16 and K17 through Nrf2-dependent and independent signaling in mouse keratinocytes [70]. In addition, SF-mediated Nrf2 signaling rescued the epidermolysis bullosa simplex (EBS), which is typified by the dysfunction of intermediated filaments in the basal keratinocytes of the epidermis [71]. Similar to the SF treatment strategy, we believe that the application of gene induction and gene repression through FA will be useful for the development of therapeutic agents for wound healing and keratinopathies via modulation of β-catenin and Nrf2, with minimal side effects.

## Figures and Tables

**Figure 1 biomedicines-09-00459-f001:**
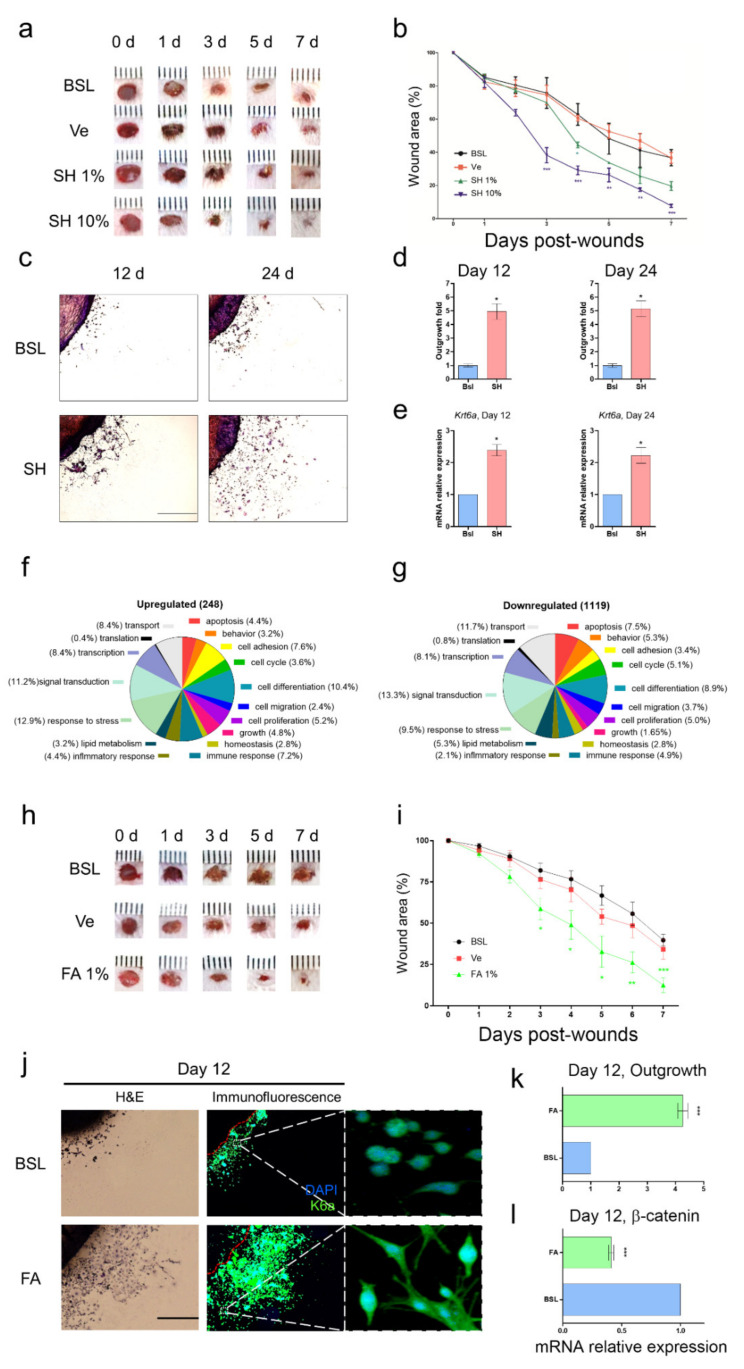
Wound healing of SH and FA. (**a**) Representative digital images of acute wounds treated with SH on days 0, 1, 3, 5, and 7 post-wounding. (**b**) Closure of acute wounds with SH was presented as percentage of wound area from the initial wound size. Mean ± SEM; *n* = 8; *, *p* < 0.05; **, *p* < 0.01; ***, *p* < 0.001 compared with basal group. Statistical analysis was performed using two-way ANOVA, Bonferroni’s test, or paired *t*-test. (**c**) Representative data of mouse skin explants processed for hematoxylin and eosin (H&E) staining at day 12 and day 24 in skin explant culture. The skin explant culture was analyzed in triplicate. (**d**) The quantitation of cellular outgrowth from explants was enhanced with SH. (**e**) The cells derived from skin explants were analyzed for Krt6α using qPCR at day 12 and day 24. Mean ± SEM; *n* = 6; *, *p* < 0.05. The significance of the outgrowth area was calculated using ImageJ software and analyzed using a paired *t*-test. (**f**,**g**) The classification criteria for the 17 categories were provided in the GO ontology databases (http://www.geneontology.org). (**h**) Representative digital images of acute wounds treated with SH on days 0, 1, 3, 5, and 7 post-wounding. (**i**) Closure of acute wounds with FA was presented as percentage of wound area from the initial wound size. Mean ± SEM; *n* = 8; *, *p* < 0.05; **, *p* < 0.01; ***, *p* < 0.001 compared with basal group. Statistical analysis was performed using two-way ANOVA, Bonferroni’s test, or paired *t*-test. BSL: basal, Ve: vehicle (poly petroleum jelly), FA: ferulic acid,%: (*w*/*w*). (**j**) Examples of mouse skin explants processed for hematoxylin and eosin (H&E) staining and immunofluorescence analyses of K6β after day 12 in ex vivo skin explant culture. (**k**) The quantitation of cell outgrowth from explants was enhanced with FA. The skin explant culture was analyzed in triplicate. Mean ± SEM; *n* = 6; ***, *p* < 0.001. The significance of outgrowth was calculated using ImageJ software and analyzed using paired *t*-test. (**l**) The cells derived from the ex vivo skin explant culture were analyzed for β-catenin using qPCR at day 12. Mean ± SEM; *n* = 6; ***, *p* < 0.001. The significance was analyzed using paired *t*-test.

**Figure 2 biomedicines-09-00459-f002:**
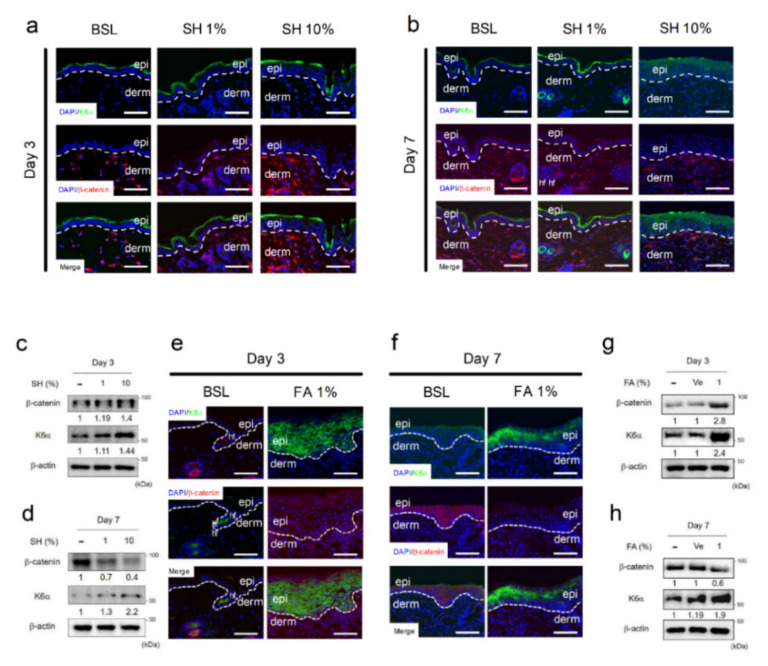
SH and FA induce epithelialization of keratinocytes. (**a**,**b**) Immunofluorescence (IF) staining of K6α and β-catenin in the epithelium at day 3 or 7 after wounding with or without SH. (**c**,**d**) Protein expression of K6α and β-catenin was analyzed after exposure to SH using Western blotting at day 3 and day 7. (**e**,**f**) K6α and β-catenin in the wounded epithelium at day 3 or 7 after exposure to FA. (**g**,**h**) Protein expression of K6α and β-catenin was analyzed after exposure to FA using Western blotting at day 3 and day 7. The IF and Western blots were analyzed in triplicate. Mean ± SEM; *n* = 5; epi: epidermis; derm: dermis.

**Figure 3 biomedicines-09-00459-f003:**
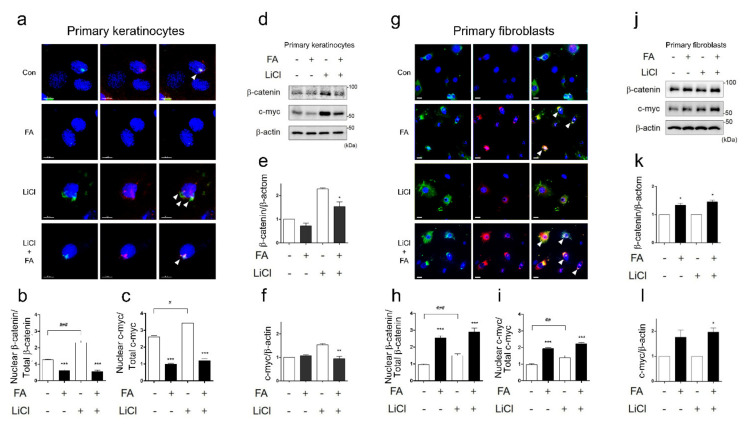
FA suppresses nucleus translocation of β-catenin in primary keratinocytes but induces nucleus translocation of β-catenin in primary fibroblasts. (**a**) After treatment with or without LiCl, immunofluorescence (IF) staining of c-myc and β-catenin was examined in primary keratinocytes. (**b**) The graph indicates the nucleus translocation ratio (%) of β-catenin. (**c**) The graph indicates the nucleus translocation ratio (%) of c-myc. Significance was measured using *t*-tests. Note: ** *p* < 0.01, scale bar: 10μm. The IF was analyzed in triplicate. Mean ± SEM; *n* = 100. (**d**) Protein expression of c-myc or β-catenin was analyzed using Western blotting in primary keratinocytes. Significance was measured using *t*-tests. Note: ** *p*<0.01. (**e**,**f**) Protein expression of β-catenin and c-myc was analyzed using Western blotting in primary keratinocytes. The Western blots were analyzed in triplicate. Mean ± SEM; *n* = 5. Significance was measured using *t*-tests. Note: * *p* < 0.05, ** *p* < 0.01. (**g**) After treatment with or without LiCl, immunofluorescence (IF) staining of c-myc and β-catenin was examined in primary fibroblasts. (**h**) The graph indicates the nucleus translocation ratio (%) of β-catenin in primary fibroblasts. (**i**) The graph indicates the nucleus translocation ratio (%) of c-myc in primary fibroblasts. Significance was measured using *t*-tests. Note: ** *p* < 0.01, scale bar: 30μm. The IF was analyzed in triplicate. Mean ± SEM; *n* = 100. (**j**) Protein expression of c-myc or β-catenin was analyzed using Western blotting in primary fibroblasts. (**k**,**l**) Protein expression of β-catenin and c-myc was analyzed using Western blotting in primary fibroblasts. The IF and Western blot samples were analyzed in triplicate. Mean ± SEM; *n* = 5. Significance was measured using *t*-tests. Note: * *p* < 0.05.

**Figure 4 biomedicines-09-00459-f004:**
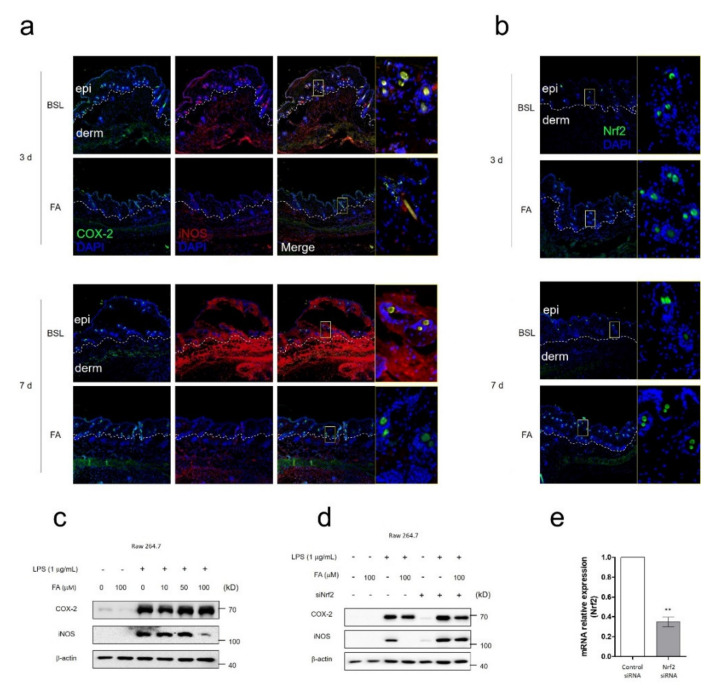
FA has anti-inflammatory effects via activation of NRF2. (**a**) Immunofluorescence (IF) staining of COX-2 and iNOS indicated in the epithelium at day 3 or 7 after wounding with or without FA. (**b**) Immunofluorescence (IF) staining of Nrf2 indicated in the epithelium at day 3 or 7 after wounding with or without FA. (**c**) Protein expression of COX-2 and iNOS was analyzed after exposure to FA using Western blotting in RAW 264.7 cells. (**d**) K6α and β-catenin are indicated in the wounded epithelium at day 3 or 7 after exposure to FA. COX-2 and iNOS expression were observed using Western blotting under Nfe2l2 knockdown conditions. (**e**) The graph showed the knockdown ratio of Nfe2l2. The IF, Western blotand Nfe2l2 knockdown test were performed in triplicate. Mean ± SEM; ** *p* < 0.01, *n* = 5. epi: epidermis, derm: dermis.

**Figure 5 biomedicines-09-00459-f005:**
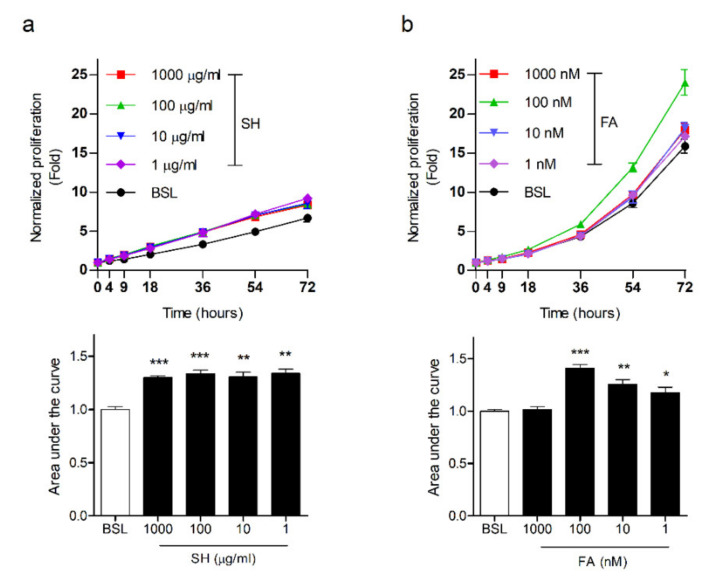
SH and FA enhance cellular proliferation. (**a**) The curves represent the course of the murine keratinocytes’ (kera 308 cell line) viability with SH for 72 h. (**b**) FA’s effects were assessed for 72 h. The RTCA was analyzed in triplicate. Each significance was measured using one-way ANOVA; *n* = 3, *** *p* < 0.001, ** *p* < 0.01, C/W; cells per well, arrow, treated point; BSL, basal; SH, shilajit; FA, ferulic acid. The graphs treated with SH or FA were compared with the area under curve using ImageJ software. (**c**) Immunofluorescence (IF) staining of Ki67 indicated in the wound epithelium at day 7. Significance was measured using *t*-tests. The IF was analyzed in triplicate. Mean ± SEM; *n* = 5, * *p*< 0.05, ** *p* < 0.01.

**Table 1 biomedicines-09-00459-t001:** Up-regulation of genes associated with wound induced keratins.

Probe Name	Description	Gene Symbol	Fold Change ^#^
**Keratins**			
1422784_at	keratin 6A	*Krt6a*	14.6901
1423227_at	keratin 17	*Krt17*	7.9747
1424096_at	keratin 5	*Krt5*	7.3153
1423935_x_at	keratin 14	*Krt14*	8.1810

^#^ Fold change: BSL/SH normalized intensity ratio.

**Table 2 biomedicines-09-00459-t002:** Alteration of genes associated with extracellular matrix.

Probe Name	Description	Gene Symbol	Fold Change ^#^
**Collagen**			
1455494_at	collagen, type I, alpha 1	*Col1a1*	2.5030
1423110_at	collagen, type I, alpha 2	*Col1a2*	2.6747
1427884_at	collagen, type III, alpha 1	*Col3a1*	3.4892
1418799_a_at	collagen, type XVII, alpha 1	*Col17a1*	2.4751
**Tight junction**			
1437932_a_at	claudin 1	*Cldn1*	2.1909
1434651_a_at	claudin 3	*Cldn3*	2.0510
**Adherens junction**			
1448261_at	cadherin 1	*Cdh1*	2.2079
**Desmosome**			
1435494_s_at	desmoplakin	*Dsp*	4.6570
1434534_at	desmocollin 3	*Dsc3*	2.1909
**Gap junction**			
1415801_at	gap junction protein, alpha 1	*Gja1*	−0.39884

^#^ Fold change: BSL/SH normalized intensity ratio.

**Table 3 biomedicines-09-00459-t003:** Alteration of genes associated with migration.

Probe Name	Description	Gene Symbol	Fold Change ^#^
**Oncogene**			
1423240_at	Rous sarcoma oncogene	*Src*	2.1337
**Integrin linked kinases**			
1449942_a_at	Integrin-linked kinase	*Ilk*	2.0405
**Chemokine ligand**			
1417574_at	chemokine (C-X-C motif) ligand 12	*Cxcl12*	3.078
**Fibroblast growth factor**			
1422916_at	fibroblast growth factor 21	*Fgf21*	2.0611
**A disintegrin and metallopeptidase**			
1450105_at	a disintegrin and metallopeptidase domain 10	*Adam10*	−0.2529
1421857_at	a disintegrin and metallopeptidase domain 17	*Adam17*	−0.1847
**Wnt signaling**			
1448818_at	wingless-related MMTV integration site 5A	*Wnt5a*	−0.4180
1430533_a_at	catenin (cadherin associated protein), beta 1	*Ctnnb1*	−0.0786

^#^ Fold change: BSL/SH normalized intensity ratio.

## Data Availability

All data that support the findings of this study are available from the corresponding author, upon reasonable request.

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
