# Peer review of "Ferulic Acid Induces Keratin 6α via Inhibition of Nuclear β-Catenin Accumulation and Activation of Nrf2 in Wound-Induced Inflammation"

_biomedicines, 2021, doi:10.3390/biomedicines9050459_

Round 1
Reviewer 1 Report
In the manuscript submitted by Kim et al., the authors demonstrated that shilajit and one of its components ferulic acid promote wound healing and migration of keratinocytes in in vivo and ex vivo models. The authors demonstrated that the pharmacological action seems to be mediated by the modulation of Keratin 6α, the activation of nuclear factor erythroid-2-related factor and the inhibition of nuclear β-catenin accumulation.
The study shows novelty, has physiological relevance and support the use of ferulic acid as natural molecule with wound healing activity.
However, there is a major concern regarding the organization of the work, the quality of the description in some sections and the presentation of scientific data obtained from the study.
The paper is not well written and there are a lot of defects that required an extensive linguistic revision. In this form, many scientific data are not clear and are not in line with the standard requested. I think that a substantial improvement for the English part is needed before considering the paper for its publication in this Journal.
Furthermore, there are also different scientific points that need to be clarified or that should be improved:
- Introduction - line 47-63: please improve this section providing a better description of the WNT beta-catenin pathway (also mentioning the canonical beta catenin-dependent signaling and non-canonical beta-catenin-independent signaling pathways) and its involvement in the wound healing process;
- The authors reported that K6, K16 and K17 are essential in the wound healing process and also in other keratin disorders as you mentioned in the conclusions. Please, explain why you do not evaluate the potential effect of SH and FA also on the expression of K16 and K17 through qPCR, immunofluorescence and western blotting.
- One of the aims of the paper, as reported by the authors, was the identification of the mechanism of action and the bioactive compounds of SH since they are still unknown. The methodology applied for the identification of FA as the main component responsible of the wound healing activity of SH is not clear. Please explain why you do not perform a bio-guided study in order to understand if also other molecules that are known to be part of SH may contribute to the wound healing effect (for example humic acid, gallic acid that are known to have wound healing activity). They may also be more active than FA.
- Relating to the last point, please explain why you do not include both SA and FA in the same in vivo study in order to compare the wound healing activity and also obtain information about other molecules that may be responsible of the pharmacological effect observed.
- Please discuss if SH could be considered more active than FA as seems observing the pharmacological effect reported in figure 1b, 1d and 1i, 1k;
- Please explain the scientific rationale that support the selection of the concentration of FA (1%) used in the in vivo study
- The authors propose the SH and FA as safe therapeutic option with minimal side effects or with a better safety profile than other approved or experimental treatment used for wound healing and other keratin disorders. This conclusion is not supported by experimental data. Please explain.
- I would also highlight that side effect that you mentioned about VA are related to systemic and not topical administration of VA and also for another therapeutic indication.
- Lines 97 “Shilajit was isolated?” or was extracted? Which type of extract you use for the chromatographic and pharmacological studies?
Other points:
- Add reference at the end of the row 36 - “Acute wounds….South America
- The sentences on lines 22-23 “Because, beta catenin….keratinocytes” are not clear and should be rephrased.
- The sentences on lines 94-95 about Shilajit is not clear. It was purchased or collected by your team?
- Line 98 “Ferulic acid purchased” is not correct. - “ Ferulic acid was purchased”
- The sentence on line 22-23 “Because, beta catenin….keratinocytes” are not clear and should be rephrased.
- The sentence on line 101-102 “Cell culture…..described” is not clear and should be rephrased.
- Line 110-111: “Cell culture were carried out? or were incubated? Same on line 116.
- The sentence on line 124 “Total RNA….keratinocytes” is not clear and should be rephrased.
- Paragraph 2.7 you mentioned only SH in the in vivo study and not FA. Please modify.
- The sentences on ines 260-262 “After sacrifice…..immunofluorescence” are not clear and should be rephrased.
- The sentences on lines 264-266 “Mazzalupo et al….culture” are not clear and should be rephrased.
- The sentences on lines 295-299 “Although FA…..components” are not clear and should be rephrased.
- The sentences on lines 486-488 “These quantitative……active compound” are not clear and should be rephrased
- The sentences on lines 514-520 “ Recently……researched” are not clear and should be rephrased.
- The sentence on line 571-572 “ In the present……fibroblasts” is not clear and should be rephrased.
- Please check correctly when you mention Figure and Supplementary Figure: line 344 - Figure 1J is not correct, it should be 1l; line 348 - Figure 1G is not correct, it should be 1i; line 302 – Supplementary figure 5 A,B,C and Supplementary figure 6 are not correct; it should be Supplementary figure 4? The same on line 600.
Author Response
Biomedicine-1175382
Reviewer 1
- Introduction - line 47-63: please improve this section providing a better description of the WNT beta-catenin pathway (also mentioning the canonical beta catenin-dependent signaling and non-canonical beta-catenin-independent signaling pathways) and its involvement in the wound healing process;
- Thanks for your comments. As reviewer 1 mentioned, we added academic explanation about WNT beta-catenin pathway. Especially, we focused on addition of canonical beta catenin-dependent signaling, because our results related with canonical beta catenin-dependent signaling during wound process.
- The authors reported that K6, K16 and K17 are essential in the wound healing process and also in other keratin disorders as you mentioned in the conclusions. Please, explain why you do not evaluate the potential effect of SH and FA also on the expression of K16 and K17 through qPCR, immunofluorescence and western blotting.
- According to evaluation of SH effects, we presented up-regulated keratin genes using microarray. Additionally, we newly uploaded qPCR results about keratin genes such as Krt6a, Krt6b, and Krt16 and Nrf2-mediated reductase gene, Nqo1 in kera-308 cell line (Supplementary figure. 7).
- One of the aims of the paper, as reported by the authors, was the identification of the mechanism of action and the bioactive compounds of SH since they are still unknown. The methodology applied for the identification of FA as the main component responsible of the wound healing activity of SH is not clear. Please explain why you do not perform a bio-guided study in order to understand if also other molecules that are known to be part of SH may contribute to the wound healing effect (for example humic acid, gallic acid that are known to have wound healing activity). They may also be more active than FA.
- In our SH sample, we could not be identified about humic acid and gallic acid using LC-MS/MS (Supplementary figure. 5). However, we detected ferulic acid and identified standardization using LC-MS/MS (Supplementary figure. 6). Finally, we evaluated that FA had more effective wound healing than SH treatment (Fig. 1 to Fig. 5).
- Relating to the last point, please explain why you do not include both SA and FA in the same in vivo study in order to compare the wound healing activity and also obtain information about other molecules that may be responsible of the pharmacological effect observed.
- Thanks for your critical comments. Even if we identified fourteen chemicals in SH sample using LC-MS/MS (Supplementary figure. 5), we could not isolate minor molecules using preparative liquid HPLC system. Because, minor molecules were slightly existences and were technically difficulty getting to isolation as much as in vivo test. Thus, we selected ferulic acid, which had detectable quantity among SH sample, and analyzed standardized component analysis using LC-MS/MS (Supplementary figure. 6).
- Please discuss if SH could be considered more active than FA as seems observing the pharmacological effect reported in figure 1b, 1d and 1i, 1k;
- According to pharmacological effect of keratin gene regulation in fig. 1d, we uploaded qPCR results about keratin genes such as Krt6a, Krt6b, and Krt16 (Supplementary figure. 7). The F 1b will be comparable with Fig. 1i under wound closure effect. The fig. 1k will be relatively comparable with catenin gene, Ctnnb1 in Table 3.
- Please explain the scientific rationale that support the selection of the concentration of FA (1%) used in the in vivo study
- To define concentration of FA in wound healing effects, we firstly screened three dose group (FA 0.1, 1, and 10%, data not shown) of FA with concentration of SH group. In our results, FA 10% was similar to 1% in wound closure assay. In addition, FA 0.1 % was similar to vehicle group in wound closure assay. In these reasons, we just presented distinguished concentration of FA 1%. In supplementary figure 4, we exhibited that FA presented remarkable healing effect in histological analysis of H&E staining after acute wound.
- The authors propose the SH and FA as safe therapeutic option with minimal side effects or with a better safety profile than other approved or experimental treatment used for wound healing and other keratin disorders. This conclusion is not supported by experimental data. Please explain.
- In our manuscript, Discussion part included in conclusion part of biological implication. Thus, we revised merge version of manuscript both Discussion and conclusion.
- I would also highlight that side effect that you mentioned about VA are related to systemic and not topical administration of VA and also for another therapeutic indication.
- Thanks for your comments. I agree your opinion. As reviewer mentioned, VA had minimal side effect in topical therapy such as tissue injury. Thus, we decided to delete the context about side effects of systemic therapeutic cases of VA.
- Lines 97 “Shilajit was isolated?” or was extracted? Which type of extract you use for the chromatographic and pharmacological studies?
- As reviewer mentioned, shilajit was extracted for qualitative analysis using LC-ESI/MS. To avoid academic confusion, we revised description for ‘For qualitative analysis using LC-ESI/MS…condition.
- In pharmacological study, we only used water extraction of SH.
Other points:
- Add reference at the end of the row 36 - “Acute wounds….South America
- We revised.
- The sentences on lines 22-23 “Because, beta catenin….keratinocytes” are not clear and should be rephrased.
- We revised bellow.
- Because, accumulation of b-catenin into nucleus acts as a negative regulator and disturbs migration in human epidermal keratinocytes.
- The sentences on lines 94-95 about Shilajit is not clear. It was purchased or collected by your team?
- We purchased SH and revised manuscripts.
- Line 98 “Ferulic acid purchased” is not correct.
- We revised.
- The sentence on line 22-23 “Because, beta catenin….keratinocytes” are not clear and should be rephrased.
- As reviewer mentioned at No. 2, we revised.
- The sentence on line 101-102 “Cell culture…..described” is not clear and should be rephrased.
- We revised.
- Line 110-111: “Cell culture were carried out? or were incubated? Same on line 116.
- We revised.
- The sentence on line 124 “Total RNA….keratinocytes” is not clear and should be rephrased.
- We revised.
- Paragraph 2.7 you mentioned only SH in the in vivo study and not FA. Please modify.
- We revised.
- The sentences on ines 260-262 “After sacrifice…..immunofluorescence” are not clear and should be rephrased.
- We revised.
- The sentences on lines 264-266 “Mazzalupo et al….culture” are not clear and should be rephrased.
- We revised.
- The sentences on lines 295-299 “Although FA…..components” are not clear and should be rephrased.
- We revised.
- The sentences on lines 486-488 “These quantitative……active compound” are not clear and should be rephrased
- We revised.
- The sentences on lines 514-520 528-534“ Recently……researched” are not clear and should be rephrased.
- We revised.
- The sentence on line 571-572 “ In the present……fibroblasts” is not clear and should be rephrased.
- We revised.
- Please check correctly when you mention Figure and Supplementary Figure:
line 344 - Figure 1J is not correct, it should be 1l;
- It was collect, and I clearly wrote explanation about ‘ex vivo’.
line 348 - Figure 1G is not correct, it should be 1i;
- It was collect, and I clearly wrote explanation about ‘ex vivo’.
line 302 – Supplementary figure 5 A,B,C and Supplementary figure 6 are not correct; it should be Supplementary figure 4? The same on line 600.
- We newly uploaded the revised format of supplementary figures.
Reviewer 2 Report
The authors wrote a paper titled Ferulic acid induces keratin 6alpha by inhibition of nuclear beta-catenin accumulation and activation of Nrf2 on wound-induced inflammation. The theme is of high importance, since there has been problem with difficult-to-heal skin wounds. The paper also contains numerous methods.
I have some comments and questions:
Is shilajit standarized?
What is basal level?
Fig. 1-5 are unclear, the titles of axes are too small.
Did you do shilajit and ferulic acid also together?
Author Response
Reviewer 2
I have some comments and questions:
Is shilajit standarized?
- We purchased shilajit in PRIMAVIE
What is basal level?
Basal means control level
- Right. Basal means control level.
Fig. 1-5 are unclear, the titles of axes are too small.
- We revised about the titles of axes.
Did you do shilajit and ferulic acid also together?
- We did not test group of co-treatment both shilajit and ferulic acid
Round 2
Reviewer 1 Report
Thanks for your revision. The authors made all the relevant corrections that I suggested.
However, before publishing, check the following points:
- I don’t see in the folder the supplementary figures 5,6,7. Please check correctly that you have uploaded the correct file. Moreover, please check correctly when you mention Supplementary Figure 5 A,B,C and Supplementary figure 6 on lines 308-310. The same on line 611-614. The supplementary figure 4 in my folder is related to the chromatographic studies.
- Please add in the text, when relevant, that aqueous extract was used in the pharmacologic studies.
Author Response
Biomedicine-1175382
To. Biomedicine editors,
Thanks for your advisory and academic comments.
As reviewer comments, we revised the manuscript.
If you have any questions, please let us know.
Best,
Hyeung-Jin Jang
Reviewer 1
Thanks for your revision. The authors made all the relevant corrections that I suggested.
However, before publishing, check the following points:
I don’t see in the folder the supplementary figures 5,6,7. Please check correctly that you have
uploaded the correct file. Moreover, please check correctly when you mention Supplementary Figure
5 A,B,C and Supplementary figure 6 on lines 308-310. The same on line 611-614. The supplementary
figure 4 in my folder is related to the chromatographic studies.
(Response) Thanks for your comments. We corrected.
Please add in the text, when relevant, that aqueous extract was used in the pharmacologic studies.
(Response) Thanks. We added.
